# Stabilizing Gradients for Deep Neural Networks via Efficient SVD Parameterization

## Abstract

Vanishing and exploding gradients are two of the main obstacles in training deep neural networks, especially in capturing long range dependencies in recurrent neural networks (RNNs). In this paper, we present an efficient parametrization of the transition matrix of an RNN that allows us to stabilize the gradients that arise in its training. Specifically, we parameterize the transition matrix by its singular value decomposition (SVD), which allows us to explicitly track and control its singular values. We attain efficiency by using tools that are common in numerical linear algebra, namely Householder reflectors for representing the orthogonal matrices that arise in the SVD. By explicitly controlling the singular values, our proposed svdRNN method allows us to easily solve the exploding gradient problem and we observe that it empirically solves the vanishing gradient issue to a large extent. We note that the SVD parameterization can be used for any rectangular weight matrix, hence it can be easily extended to any deep neural network, such as a multi-layer perceptron. Theoretically, we demonstrate that our parameterization does not lose any expressive power, and show how it potentially makes the optimization process easier. Our extensive experimental results also demonstrate that the proposed framework converges faster, and has good generalization, especially in capturing long range dependencies, as shown on the synthetic addition and copying tasks.

## 1 Introduction

Deep neural networks have achieved great success in various fields, including computer vision, speech recognition, natural language processing, etc. Despite their tremendous capacity to fit complex functions, optimizing deep neural networks remains a contemporary challenge. Two main obstacles are vanishing and exploding gradients, that become particularly problematic in Recurrent Neural Networks (RNNs) since the transition matrix is identical at each layer, and any slight change to it is amplified through recurrent layers (Bengio et al. (1994)).

Several methods have been proposed to solve the issue, for example, Long Short Term Memory (LSTM) (Hochreiter & Schmidhuber (1997)) and residual networks (He et al. (2016)). Another recently proposed class of methods is designed to enforce orthogonality of the square transition matrices, such as unitary and orthogonal RNNs (oRNN) (Arjovsky et al. (2016); Mhammedi et al. (2017)). However, while these methods solve the exploding gradient problem, they limit the expressivity of the network.

In this paper, we present an efficient parametrization of weight matrices that arise in a deep neural network, thus allowing us to stabilize the gradients that arise in its training, while retaining the desired expressive power of the network. In more detail we make the following contributions:

- We propose a method to parameterize weight matrices through their singular value decomposition (SVD). Inspired by (Mhammedi et al. (2017)), we attain efficiency by using tools that are common in numerical linear algebra, namely Householder reflectors for representing the orthogonal matrices that arise in the SVD. The SVD parametrization allows us to retain the desired expressive power of the network, while enabling us to explicitly track and control singular values.
- We apply our SVD parameterization to recurrent neural networks to exert spectral constraints on the RNN transition matrix. Our proposed svdRNN method enjoys similar space and time complexity as the vanilla RNN. We empirically verify the superiority of svdRNN over RNN/oRNN, in some case even LSTMs, over an exhaustive collection of time series classification tasks and the synthetic addition and copying tasks, especially when the network depth is large.

- Theoretically, we show how our proposed SVD parametrization can make the optimization process easier. Specifically, under a simple setting, we show that there are no spurious local minimum for the linear svdRNN in the population risk.

- Our parameterization is general enough to eliminate the gradient vanishing/exploding problem not only in RNNs, but also in various deep networks. We illustrate this by applying SVD parametrization to problems with non-square weight matrices, specifically multi-layer perceptrons (MLPs) and residual networks.

We now present the outline of our paper. In Section 2, we discuss related work, while in Section 3 we introduce our SVD parametrization and demonstrate how it spans the whole parameter space and does not limit expressivity. In Section 4 we propose the svdRNN model that is able to efficiently control and track the singular values of the transition matrices, and we extend our parameterization to non-square weight matrices and apply it to MLPs in Section 5. Section 6 provides the optimization landscape of svdRNN by showing that linear svdRNN has no spurious local minimum. Experimental results on MNIST and a popular time series archive are present in Section 7. Finally, we present our conclusions and future work in Section 8.

## 2 RELATED WORK

Numerous approaches have been proposed to address the vanishing and exploding gradient problem. Long short-term memory (LSTM) (Hochreiter & Schmidhuber (1997)) attempts to address the vanishing gradient problem by adding additional memory gates. Residual networks (He et al. (2016)) pass the original input directly to the next layer in addition to the original layer output. Mikolov (2012) performs gradient clipping, while Pascanu et al. (2013) applies spectral regularization to the weight matrices. Other approaches include introducing $L_1$ or $L_2$ penalization on successive gradient norm pairs in back propagation (Pascanu et al. (2013)).

Recently the idea of restricting transition matrices to be orthogonal has drawn some attention. Le et al. (2015) proposed initializing recurrent transition matrices to be identity or orthogonal (IRNN). This strategy shows better performance when compared to vanilla RNN and LSTM. However, there is no guarantee that the transition matrix is close to orthogonal after a few iterations. The unitary RNN (uRNN) algorithm proposed in Arjovsky et al. (2016) parameterizes the transition matrix with reflection, diagonal and Fourier transform matrices. By construction, uRNN ensures that the transition matrix is unitary at all times. Although this algorithm performs well on several small tasks, Wisdom et al. (2016) showed that uRNN only covers a subset of possible unitary matrices and thus detracts from the expressive power of RNN. An improvement over uRNN, the orthogonal RNN (oRNN), was proposed by Mhammedi et al. (2017). oRNN uses products of Householder reflectors to represent an orthogonal transition matrix, which is rich enough to span the entire space of orthogonal matrices. Meanwhile, Vorontsov et al. (2017) empirically demonstrate that the strong constraint of orthogonality limits the model's expressivity, thereby hindering its performance. Therefore, they parameterize the transition matrix by its SVD, $W = U\Sigma V^\top$ (factorized RNN) and restrict $\Sigma$ to be in a range close to 1; however, the orthogonal matrices $U$ and $V$ are updated by geodesic gradient descent using the Cayley transform, thereby resulting in time complexity cubic in the number of hidden nodes which is prohibitive for large scale problems. Motivated by the shortcomings of the above methods, our work in this paper attempts to answer the following questions: *Is there an efficient way to solve the gradient vanishing/exploding problem without hurting expressive power?*

*As brought to wide notice in He et al. (2016), deep neural networks should be able to preserve features that are already good. Hardt & Ma (2016) consolidate this point by showing that deep linear residual networks have no spurious local optima. In our work, we broaden this concept and bring it to the area of recurrent neural networks, showing that each layer is not necessarily near identity, but being close to orthogonality suffices to get a similar result.*

*Generalization is a major concern in training deep neural networks. Bartlett et al. (2017) provide a generalization bound for neural networks by a spectral Lipschitz constant, namely the product of spectral norm of each layer. Thus, our scheme of restricting the spectral norm of weight matrices reduces generalization error in the setting of Bartlett et al. (2017). As supported by the analysis in Cisse et al. (2017), since our SVD parametrization allows us to develop an efficient way to constrain the weight matrix to be a tight frame (Tropp et al. (2005)), we consequently are able to reduce the sensitivity of the network to adversarial examples.*

## 3 SVD PARAMETERIZATION

*The SVD of the transition matrix $W \in \mathbb{R}^{n \times n}$ of an RNN is given by $W = U\Sigma V^T$, where $\Sigma$ is the diagonal matrix of singular values, and $U, V \in \mathbb{R}^{n \times n}$ are orthogonal matrices, i.e., $U^T U = UU^T = I$ and $V^T V = VV^T = I$ (Trefethen & Bau III (1997)). During the training of an RNN, our proposal is to maintain the transition matrix in its SVD form. However, in order to do so efficiently, we need to maintain the orthogonal matrices $U$ and $V$ in compact form, so that they can be easily updated by forward and backward propagation. In order to do so, as in Mhammedi et al. (2017), we use a tool that is commonly used in numerical linear algebra, namely Householder reflectors (which, for example, are used in computing the $QR$ decomposition of a matrix).*

*Given a vector $u \in \mathbb{R}^k, k \leq n$, the $n \times n$ Householder reflector $\mathcal{H}_k^n(u)$ is defined as:*

$$\mathcal{H}_k^n(u) = \begin{cases} \begin{pmatrix} I_{n-k} & \\ & I_k - 2\frac{uu^\top}{\|u\|^2} \end{pmatrix} & , \quad u \neq \mathbf{0} \\ I_n & , \quad \text{otherwise.} \end{cases} \tag{1}$$

*The Householder reflector is clearly a symmetric matrix, and it can be shown that it is orthogonal, i.e., $H^2 = I$ (Householder (1958)). Further, when $u \neq 0$, it has $n-1$ eigenvalues that are 1, and one eigenvalue which is $-1$ (hence the name that it is a reflector) . In practice, to store a Householder reflector, we only need to store $u \in \mathbb{R}^k$ rather than the full matrix.*

*Given a series of vectors $\{u_i\}_{i=k}^n$ where $u_k \in \mathbb{R}^k$, we define the map:*

$$\mathcal{M}_k : \mathbb{R}^k \times ... \times \mathbb{R}^n \mapsto \mathbb{R}^{n \times n}$$
$$(u_k, ..., u_n) \mapsto \mathcal{H}_n(u_n)...\mathcal{H}_k(u_k), \tag{2}$$

*where the right hand side is a product of Householder reflectors, yielding an orthogonal matrix (to make the notation less cumbersome, we remove the superscript from $\mathcal{H}_k^n$ for the rest of this section).*

**Theorem 1.** *The image of $\mathcal{M}_1$ is the set of all $n \times n$ orthogonal matrices.*

*The proof of Theorem 1 is an easy extension of the Householder QR factorization Theorem, and is presented in Appendix A. Although we cannot express all $n \times n$ matrices with $\mathcal{M}_k$, any $W \in \mathbb{R}^{n \times n}$ can be expressed as the product of two orthogonal matrices $U, V$ and a diagonal matrix $\Sigma$, i.e. by its SVD: $W = U\Sigma V^\top$. Given $\sigma \in \mathbb{R}^n$ and $\{u_i\}_{i=k_1}^n, \{v_i\}_{i=k_2}^n$ with $u_i, v_i \in \mathbb{R}^i$, we finally define our proposed SVD parametrization:*

$$\mathcal{M}_{k_1,k_2} : \mathbb{R}^{k_1} \times ... \times \mathbb{R}^n \times \mathbb{R}^{k_2} \times ... \times \mathbb{R}^n \times \mathbb{R}^n \mapsto \mathbb{R}^{n \times n}$$
$$(u_{k_1}, ..., u_n, v_{k_2}, ..., v_n, \sigma) \mapsto \mathcal{H}_n(u_n)...\mathcal{H}_{k_1}(u_{k_1})diag(\sigma)\mathcal{H}_{k_2}(v_{k_2})...\mathcal{H}_n(v_n). \tag{3}$$

**Theorem 2.** *The image of $\mathcal{M}_{1,1}$ is the set of $n \times n$ real matrices.*
*i.e. $\mathbb{R}^{n \times n} = \mathcal{M}_{1,1}\left(\mathbb{R}^1 \times ... \times \mathbb{R}^n \times \mathbb{R}^1 \times ... \times \mathbb{R}^n \times \mathbb{R}^n\right)$*

*The proof of Theorem 2 is based on the singular value decomposition and Theorem 1, and is presented in Appendix A. The astute reader might note that $\mathcal{M}_{1,1}$ seemingly maps an input space of $n^2 + 2n$ dimensions to a space of $n^2$ dimensions; however, since $\mathcal{H}_k^n(u_k)$ is invariant to the norm of $u_k$, the input space also has exactly $n^2$ dimensions. Although Theorems 1 and 2 are simple extensions of well known linear algebra results, they ensure that our parameterization has the ability to represent any matrix and so the full expressive power of the RNN is retained.*

**Theorem 3.** *The image of $\mathcal{M}_{k_1,k_2}$ includes the set of all orthogonal $n \times n$ matrices if $k_1 + k_2 \leq n+2$.*

*Theorem 3 indicates that if the total number of reflectors is greater than $n$: $(n-k_1+1)+(n-k_2+1) \geq n$, then the parameterization covers all orthogonal matrices. Note that when fixing $\sigma = \mathbf{1}$, $\mathcal{M}_{k_1,k_2}(\{u_i\}_{i=k_1}^n, \{v_i\}_{i=k_2}^n, \mathbf{1}) \in \mathbf{O}(n)$, where $\mathbf{O}(n)$ is the set of $n \times n$ orthogonal matrices. Thus when $k_1 + k_2 \leq n+2$, we have $\mathbf{O}(n) = \mathcal{M}_{k_1,k_2}\left[\mathbb{R}^{k_1} \times ... \times \mathbb{R}^n \times \mathbb{R}^{k_2} \times ... \times \mathbb{R}^n \times \mathbf{1}\right]$.*

## 4 SVD-RNN

*In this section, we apply our SVD parameterization to RNNs and describe the resulting svdRNN algorithm in detail. Given a hidden state vector from the previous step $h^{(t-1)} \in \mathbb{R}^n$ and input $x^{(t-1)} \in \mathbb{R}^{n_i}$, RNN computes the next hidden state $h^{(t)}$ and output vector $o^{(t)} \in \mathbb{R}^{n_o}$ as:*

$$h^{(t)} = \sigma(Wh^{(t-1)} + Mx^{(t-1)} + b) \tag{4}$$
$$o^{(t)} = Yh^{(t)} \tag{5}$$

*In svdRNN we parametrize the transition matrix $W \in \mathbb{R}^{n \times n}$ using $m_1 + m_2$ Householder reflectors as:*

$$W = \mathcal{M}_{n-m_1+1, n-m_2+1}(u_{n-m_1+1}, ..., u_n, v_{n-m_2+1}, ..., v_n, \sigma) \tag{6}$$

$$= \mathcal{H}_n(u_n)...\mathcal{H}_{n-m_1+1}(u_{n-m_1+1})diag(\sigma)\mathcal{H}_{n-m_2+1}(v_{n-m_2+1})...\mathcal{H}_n(v_n) \tag{7}$$

*This parameterization gives us several advantages over the regular RNN. First, we can select the number of reflectors $m_1$ and $m_2$ to balance expressive power versus time and space complexity. By Theorem 2, the choice $m_1 = m_2 = n$ gives us the same expressive power as vanilla RNN. Notice oRNN could be considered a special case of our parametrization, since when we set $m_1 + m_2 \geq n$ and $\sigma = \mathbf{1}$, we can represent all orthogonal matrices, as proven by Theorem 3. Most importantly, we are able to explicitly control the singular values of the transition matrix. In most cases, we want to constrain the singular values to be within a small interval near 1. The most intuitive method is to clip the singular values that are out of range. Another approach would be to initialize all singular values to 1, and add a penalty term $\|\sigma - 1\|^2$ to the objective function. Here, we have applied another parameterization of $\sigma$ proposed in Vorontsov et al. (2017):*

$$\sigma_i = 2r(f(\hat{\sigma}_i) - 0.5) + \sigma^*, \ i \in [n] \tag{8}$$

*where $f$ is the sigmoid function and $\hat{\sigma}_i$ is updated from $u_i, v_i$ via stochastic gradient descent. The above allows us to constrain $\sigma_i$ to be within $[\sigma^* - r, \sigma^* + r]$. In practice, $\sigma^*$ is usually set to 1 and $r \ll 1$. Note that we are not incurring more computation cost or memory for the parameterization. For regular RNN, the number of parameters is $(n_o + n_i + n + 1)n$, while for svdRNN it is $(n_o + n_i + m_1 + m_2 + 2)n - \frac{m_1^2 + m_2^2 - m_1 - m_2}{2}$. In the extreme case where $m_1 = m_2 = n$, it becomes $(n_o + n_i + n + 3)n$. Later we will show that the computational cost of svdRNN is also of the same order as RNN in the worst case.*

## 4.1 FORWARD/BACKWARD PROPAGATION

*In forward propagation, we need to iteratively evaluate $h^{(t)}$ from $t = 0$ to $L$ using (4). The only different aspect from a regular RNN in the forward propagation is the computation of $Wh^{(t-1)}$. Note that in svdRNN, $W$ is expressed as product of $m_1 + m_2$ Householder matrices and a diagonal matrix. Thus $Wh^{(t-1)}$ can be computed iteratively using $(m_1 + m_2)$ inner products and vector additions. Denoting $\hat{u}_k = \binom{\mathbf{0}_{n-k}}{u_k}$, we have:*

$$\mathcal{H}_k(u_k)h = \left(I_n - \frac{2\hat{u}_k\hat{u}_k^\top}{\hat{u}_k^\top \hat{u}_k}\right)h = h - 2\frac{\hat{u}_k^\top h}{\hat{u}_k^\top \hat{u}_k}\hat{u}_k \tag{9}$$

*Thus, the total cost of computing $Wh^{(t-1)}$ is $O((m_1 + m_2)n)$ floating point operations (flops). Detailed analysis can be found in Section 4.2. Let $L(\{u_i\}, \{v_i\}, \sigma, M, Y, b)$ be the loss or objective function, $C^{(t)} = Wh^{(t)}, \hat{\Sigma} = diag(\hat{\sigma})$. Given $\frac{\partial L}{\partial C^{(t)}}$, we define:*

$$\frac{\partial L}{\partial u_k^{(t)}} := \left[\frac{\partial C^{(t)}}{\partial u_k^{(t)}}\right]^\top \frac{\partial L}{\partial C^{(t)}}; \quad \frac{\partial L}{\partial v_k^{(t)}} := \left[\frac{\partial C^{(t)}}{\partial v_k^{(t)}}\right]^\top \frac{\partial L}{\partial C^{(t)}}; \tag{10}$$

$$\frac{\partial L}{\partial \Sigma^{(t)}} := \left[\frac{\partial C^{(t)}}{\partial \Sigma^{(t)}}\right]^\top \frac{\partial L}{\partial C^{(t)}}; \quad \frac{\partial L}{\partial \hat{\Sigma}^{(t)}} := \left[\frac{\partial \Sigma^{(t)}}{\partial \hat{\Sigma}^{(t)}}\right]^\top \frac{\partial L}{\partial \Sigma^{(t)}}; \tag{11}$$

$$\frac{\partial L}{\partial h^{(t-1)}} := \left[\frac{\partial C^{(t)}}{\partial h^{(t-1)}}\right]^\top \frac{\partial L}{\partial C^{(t)}} \tag{12}$$

*Back propagation for svdRNN requires $\frac{\partial C^{(t)}}{\partial u_k^{(t)}}$, $\frac{\partial C^{(t)}}{\partial v_k^{(t)}}$, $\frac{\partial C^{(t)}}{\partial \hat{\Sigma}^{(t)}}$ and $\frac{\partial C^{(t)}}{\partial h^{(t-1)}}$. These partial gradients can also be computed iteratively by computing the gradient of each Householder matrix at a time. We drop the superscript $(t)$ now for ease of exposition. Given $\hat{h} = \mathcal{H}_k(u_k)h$ and $g = \frac{\partial L}{\partial \hat{h}}$, we have*

$$\frac{\partial L}{\partial h} = \left[\frac{\partial \hat{h}}{\partial h}\right]^\top \frac{\partial L}{\partial \hat{h}} = \left(I_n - \frac{2\hat{u}_k\hat{u}_k^\top}{\hat{u}_k^\top \hat{u}_k}\right)g = g - 2\frac{\hat{u}_k^\top g}{\hat{u}_k^\top \hat{u}_k}\hat{u}_k \tag{13}$$

$$\frac{\partial L}{\partial \hat{u}_k} = \left[\frac{\partial \hat{h}}{\partial \hat{u}_k}\right]^\top \frac{\partial L}{\partial \hat{h}} = -2\left(\frac{\hat{u}_k^\top h}{\hat{u}_k^\top \hat{u}_k}I_n + \frac{1}{\hat{u}_k^\top \hat{u}_k}h\hat{u}_k^\top + \frac{\hat{u}_k^\top h}{(\hat{u}_k^\top \hat{u}_k)^2}\hat{u}_k\hat{u}_k^\top\right)g \tag{14}$$

$$= -2\frac{\hat{u}_k^\top h}{\hat{u}_k^\top \hat{u}_k}g - 2\frac{\hat{u}_k^\top g}{\hat{u}_k^\top \hat{u}_k}h + 4\frac{\hat{u}_k^\top h}{\hat{u}_k^\top \hat{u}_k}\frac{\hat{u}_k^\top g}{\hat{u}_k^\top \hat{u}_k}\hat{u}_k \tag{15}$$

*Details of forward and backward propagation can be found in Appendix (B). One thing worth noticing is that the oRNN method in Mhammedi et al. (2017) actually omitted the last term in (15) by assuming that $\|u_k\|$ are fixed. Although the scaling of $u_k$ in the Householder transform does not affect the transform itself, it does produce different gradient update for $u_k$ even if it is scaled to norm 1 afterwards.*

## 4.2 COMPLEXITY ANALYSIS

*Table 1 gives the time complexity of various algorithms. $Hprod$ and $Hgrad$ are defined in Algorithm 2 3 (see Appendix (B)). Algorithm 2 needs $6k$ flops, while Algorithm 3 uses $(3n + 10k)$ flops. Since $\|u_k\|^2$ only needs to be computed once per iteration, we can further decrease the flops to $4k$ and $(3n + 8k)$. Also, in back propagation we can reuse $\alpha$ in forward propagation to save $2k$ flops.*

|  | flops |
| --- | --- |
| $Hprod(h, u_k)$ | $4k$ |
| $Hgrad(h, u_k, g)$ | $3n + 6k$ |
| svdRNN-Local FP$(n, m_1, m_2)$ | $4n(m_1 + m_2) - 2m_1^2 - 2m_2^2 + O(n)$ |
| svdRNN-Local BP$(n, m_1, m_2)$ | $6n(m_1 + m_2) - 1.5m_1^2 - 1.5m_2^2 + O(n)$ |
| oRNN-Local FP$(n, m)$ | $4nm - m^2 + O(n)$ |
| oRNN-Local BP$(n, m)$ | $7nm - 2m^2 + O(n)$ |

Table 1: Time complexity across algorithms

## 5 EXTENDING SVD PARAMETERIZATION TO GENERAL WEIGHT MATRICES

*In this section, we extend the parameterization to non-square matrices and use Multi-Layer Perceptrons(MLP) as an example to illustrate its application to general deep networks. For any weight matrix $W \in \mathbb{R}^{m \times n}$(without loss of generality $m \leq n$), its reduced SVD can be written as:*

$$W = U(\Sigma|0)(V_L|V_R)^\top = U\Sigma V_L^\top,$$  (16)

*where $U \in \mathbb{R}^{m \times m}$, $\Sigma \in diag(\mathbb{R}^m)$, $V_L \in \mathbb{R}^{n \times m}$. There exist $u_n, ..., u_{k_1}$ and $v_n, ..., v_{k_2}$ s.t. $U = \mathcal{H}_m^m(u_m)...\mathcal{H}_{k_1}^m(u_{k_1})$, $V = \mathcal{H}_n^n(v_n)...\mathcal{H}_{k_2}^n(v_{k_2})$, where $k_1 \in [m], k_2 \in [n]$. Thus we can extend the SVD parameterization for any non-square matrix:*

$$\mathcal{M}_{k_1,k_2}^{m,n} : \mathbb{R}^{k_1} \times ... \times \mathbb{R}^m \times \mathbb{R}^{k_2} \times ... \times \mathbb{R}^n \times \mathbb{R}^{\min(m,n)} \mapsto \mathbb{R}^{m \times n}$$

$$(u_{k_1}, ..., u_m, v_{k_2}, ..., v_n, \sigma) \mapsto \mathcal{H}_m^m(u_m) \cdots \mathcal{H}_{k_1}^m(u_{k_1})\hat{\Sigma}\mathcal{H}_{k_2}^n(v_{k_2}) \cdots \mathcal{H}_n^n(v_n).$$  (17)

*where $\hat{\Sigma} = (diag(\sigma)|0)$ if $m < n$ and $(diag(\sigma)|0)^\top$ otherwise. Next we show that we only need $2\min(m, n)$ reflectors (rather than $m + n$) to parametrize any $m \times n$ matrix. By the definition of $\mathcal{H}_k^n$, we have the following lemma:*

**Lemma 1.** *Given $\{v_i\}_{i=1}^n$, define $V^{(k)} = \mathcal{H}_n^n(v_n)...\mathcal{H}_k^n(v_k)$ for $k \in [n]$. We have:*
$$V_{*,i}^{(k_1)} = V_{*,i}^{(k_2)}, \forall k_1, k_2 \in [n], i \leq \min(n - k_1, n - k_2).$$

*Here $V_{*,i}$ indicates the $i$th column of matrix $V$. According to Lemma 1, we only need at most first $m$ Householder vectors to express $V_L$, which results in the following Theorem:*

**Theorem 4.** *If $m \leq n$, the image of $\mathcal{M}_{1,n-m+1}^{m,n}$ is the set of all $m \times n$ matrices; else the image of $\mathcal{M}_{n-m+1,1}^{m,n}$ is the set of all $m \times n$ matrices.*

*Similarly if we constrain $u_i, v_i$ to have unit length, the input space dimensions of $\mathcal{M}_{1,n-m+1}^{m,n}$ and $\mathcal{M}_{m-n+1,1}^{m,n}$ are both $mn$, which matches the output dimension. Thus we extend Theorem 2 to the non-square case, which enables us to apply SVD parameterization to not only the RNN transition matrix, but also to general weight matrices in various deep learning models. For example, the Multilayer perceptron (MLP) model is a class of feedforward neural network with fully connected layers:*

$$h^{(t)} = f(W^{(t-1)}h^{(t-1)} + b^{(t-1)})$$  (18)

*Here $h^{(t)} \in \mathbb{R}^{n_t}$, $h^{(t-1)} \in \mathbb{R}^{n_{t-1}}$ and $W^{(t)} \in \mathbb{R}^{n_t \times n_{t-1}}$. Applying SVD parameterization to $W^{(t)}$ say $n_t < n_{t-1}$, we have:*

$$W^{(t)} = \mathcal{H}_{n_t}^{n_t}(u_{n_t})...\mathcal{H}_1^{n_t}(u_1)\Sigma\mathcal{H}_{n_{t-1}-n_t+1}^{n_{t-1}}(v_{n_{t-1}-n_t+1})...\mathcal{H}_{n_{t-1}}^{n_{t-1}}(v_{n_{t-1}}).$$

*We can use the same forward/backward propagation algorithm as described in Algorithm 1. Besides RNN and MLP, SVD parameterization method also applies to more advanced frameworks, such as Residual networks and LSTM, which we will not describe in detail here.*

## 6 THEORETICAL ANALYSIS

*Since we can control and upper bound the singular values of the transition matrix in svdRNN, we can clearly eliminate the exploding gradient problem. In this section, we now analytically illustrate the advantages of svdRNN with lower-bounded singular values from the optimization perspective. For the theoretical analysis in this section, we will limit ourselves to a linear recurrent neural network, i.e., an RNN without any activation.*

### 6.1 REPRESENTATIONS OF RNN WITHOUT ACTIVATION

**Linear recurrent neural network.** *For simplicity, we follow a setting similar to Hardt & Ma (2016). For compact presentation, we stack the input data as $\mathcal{X} \in \mathbb{R}^{n \times t}$, where $\mathcal{X} = \left(x^{(0)}|x^{(1)}|\cdots|x^{(t-1)}\right)$, and transition weights as $\mathcal{W} \in \mathbb{R}^{n \times nt}$ where $\mathcal{W} = \left(W|W^2|\cdots|W^t\right)$. Then we can simplify the output as:*

$$o^{(t)}(\mathcal{X}) = Y(W^t h^{(0)} + \sum_{i=1}^{t} W^i(Mx^{(t)} + b))$$

*By absorbing $M$ and $b$ in each data $x^{(t)}$ and assuming $h^{(0)} = 0$, we further simplify the output as:*

$$o^{(t)}(\mathcal{X}) = Y \sum_{i=1}^{t} W^i x^{(t-1)}$$

*Suppose the input data $\mathcal{X} \sim \mathcal{D}$, and assume its underlying relation to the output is $y = A vec(\mathcal{X}) + \eta$, where $A \in \mathbb{R}^{n \times nt}$ and residue $\eta \in \mathbb{R}^n$ satisfies $\mathbb{E}_{\mathcal{X} \sim \mathcal{D}}[\eta | \mathcal{X}] = 0$. We consider the individual loss:*

$$f(W; \mathcal{X}, y) = \|o^{(t)}(\mathcal{X}) - y\|_2^2 = \|Y\mathcal{W}vec(\mathcal{X}) - y\|_2^2.$$

**Claim 1.** *With linear recurrent neural networks, the population risk*

$$R[W] = \mathbb{E}_{\mathcal{X} \sim \mathcal{D}}[f(W; \mathcal{X}, y)] = \|(Y\mathcal{W} - A)\Sigma^{1/2}\|_F^2 + C,$$

*where $\Sigma = \mathbb{E}_{\mathcal{X} \sim \mathcal{D}}[vec(\mathcal{X})vec(\mathcal{X})^\top]$, and $C = \mathbb{E}[\|\eta\|_2^2]$. Meanwhile*

$$\nabla_W R[W] = (Y\mathcal{W} - A)\Sigma \left(I_d|2W|3W^2|\cdots|tW^{t-1}\right)^\top$$

### 6.2 ALL CRITICAL POINTS ARE GLOBAL MINIMUM

**Theorem 5.** *With linear recurrent neural networks, if transition matrix $W$ satisfies $\sigma_{\min}(W) \geq e > 0$, all critical points in the population risk are global minimum.*

*Proof.* Write $Y\mathcal{W} - A$ as $(E_1|E_2|\cdots|E_t)$, where each $E_i \in \mathbb{R}^{d \times d}$. By Claim 1,

$$\|\nabla_W R[W]\|_F^2 = \|(Y\mathcal{W} - A)\Sigma \left(I_d|2W^\top|3(W^\top)^2|\cdots|t(W^\top)^{t-1}\right)^\top\|_F^2$$

$$\geq \sigma_{\min}^2(\Sigma)\|(Y\mathcal{W} - A)\left(I_d|2W^\top|3(W^\top)^2|\cdots|t(W^\top)^{t-1}\right)^\top\|_F^2$$

$$\geq \sigma_{\min}^2(\Sigma)\sum_{i=1}^{t} i^2 e^{2(i-1)}\|E_i\|_F^2$$

$$\geq \sigma_{\min}^2(\Sigma)\min_{1 \leq i \leq t}\{i^2 e^{2(i-1)}\}\|Y\mathcal{W} - A\|_F^2$$

$$\geq \sigma_{\min}^2(\Sigma)\min_{1 \leq i \leq t}\{i^2 e^{2(i-1)}\}(R(W) - R^*)$$

Therefore when $\nabla_W R[W] = 0$ suffices $R(W) = R^*$, meaning $W$ reaches the global minimum. $\square$

*Theorem 5 potentially explains why our system is easier to optimize, since with our scheme of SVD parametrization, we have the following corollary.*

**Corollary 1.** *With the update rule in* (8)*, linear svdRNNs have no spurious local minimum.*

*While the above analysis lends further credence to our observed experimental results, we leave it to future work to perform a similar analysis in the presence of non-linear activation functions.*

## 7 EXPERIMENTAL RESULTS

*In this section, we provide empirical evidence that shows the advantages of SVD parameterization in both RNNs and MLPs. For RNN models, we compare our svdRNN algorithm with (vanilla) RNN, IRNN(Le et al. (2015)), oRNN(Mhammedi et al. (2017)) and LSTM (Hochreiter & Schmidhuber (1997)). The transition matrix in IRNN is initialized to be orthogonal while other matrices are initialized by sampling from a Gaussian distribution. For MLP models, we implemented vanilla MLP, Residual Network (ResNet)(He et al. (2016)) and used SVD parameterization for both of them. We used a residual block of two layers in ResNet. In most cases leaky_Relu is used as activation*

*function, except for LSTM, where* $leaky\_Relu$ *will drastically harm the performance. To train these models, we applied Adam optimizer with stochastic gradient descent (Kingma & Ba (2014)). These models are implemented with Theano (Al-Rfou et al. (2016)).*[1]

## 7.1 TIME SERIES CLASSIFICATION

*In this experiment, we focus on the time series classification problem, where time series are fed into RNN sequentially, which then tries to predict the right class upon receiving the sequence end (Hüsken & Stagge (2003)). The dataset we choose is the largest public collection of class-labeled time-series with widely varying length, namely, the UCR time-series collection from Chen et al. (2015)*[2]. *We present the test accuracy on 20 datasets with RNN, LSTM, oRNN and svdRNN in Table 3(Appendix C) and Figure 1. In all experiments, we used hidden dimension* $n_h = 32$, *and chose total number of reflectors for oRNN and svdRNN to be* $m = 16$ *(for svdRNN* $m_1 = m_2 = 8$*). We choose proper depth* $t$ *as well as input size* $n_i$. *Given sequence length* $L$, *since* $tn_i = L$, *we choose* $n_i$ *to be the maximum divisor of* $L$ *that satisfies* $depth \leq \sqrt{L}$. *To have a fair comparison*

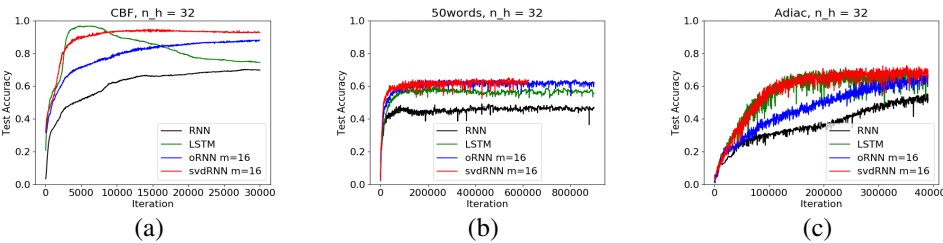

Figure 1: Performance comparisons of the RNN based models on three UCR datasets.

*of how the proposed principle itself influences the training procedure, we did not use dropout in any of these models. As illustrated in the optimization process in Figure 1, this resulted in some overfitting (see (a) CBF), but on the other hand it shows that svdRNN is able to prevent overfitting. This supports our claim that since generalization is bounded by the spectral norm of the weights Bartlett et al. (2017), svdRNN will potentially generalize better than other schemes. This phenomenon is more drastic when the depth is large (e.g. ArrowHead(251 layers) and FaceAll(131 layers)), since regular RNN, and even LSTM, have no control over the spectral norms. Also note that there are substantially fewer parameters in oRNN and svdRNN as compared to LSTM.*

## 7.2 MNIST

*In this experiment, we compare different models on the MNIST image dataset. The dataset was split into a training set of 60000 instances and a test set of 10000 instances. The* $28 \times 28$ *MNIST pixels are flattened into a vector and then traversed by the RNN models. Table 2 shows accuracy scores across multiple We tested different models with different network depth as well as width. Figure 2(a)(b) shows the test accuracy on networks with 28 and 112 layers (20 and 128 hidden dimensions) respectively. It can be seen that the svdRNN algorithms have the best performance and the choice of* $r$ *(the amount that singular values are allowed to deviate from 1) does not have much influence on the final precision. Also we explored the effect of different spectral constraints and explicitly tracked the spectral margin (*$\max_i |\sigma_i - 1|$*) of the transition matrix. Intuitively, the influence of large spectral margin should increase as the network becomes deeper. Figure 2(d) shows the spectral margin of different RNN models. Although IRNN has small spectral margin at first few iterations, it quickly deviates from orthogonal and cannot match the performance of oRNN and svdRNN. Figure 2(e) shows the magnitude of first layer gradient* $\|\frac{\partial L}{\partial h^{(0)}}\|_2$. *RNN suffers from vanishing gradient at first 50k iterations while oRNN and svdRNN are much more stable. Note that LSTM can perform relatively well even though it has exploding gradient in the first layer.*

*We also tested RNN and svdRNN with different amount of non-linearity, as shown in Figure 2(c). This is achieved by adjusting the leak parameter in* $leaky\_Relu$: $f(x) = \max(leak \cdot x, x)$. *With* $leak = 1.0$, *it reduces to the identity map and when* $leak = 0$ *we are at the original* $Relu$ *function. From the figures, we show that svdRNN is resistant to different amount of non-linearity, namely converge faster and achieve higher accuracy invariant to the amount of the* $leak$ *factor. To explore the reason underneath, we illustrate the gradient in Figure 2(f), and find out svdRNN could eliminate the gradient vanishing problem on all circumstances, while RNN suffers from gradient vanishing when non-linearity is higher.*

---

[1]we thank Mhammedi for providing their code for oRNN(Mhammedi et al. (2017))

[2]Details of the data sets, including how to split into train/vaildiation/test sets, are given in Appendix C

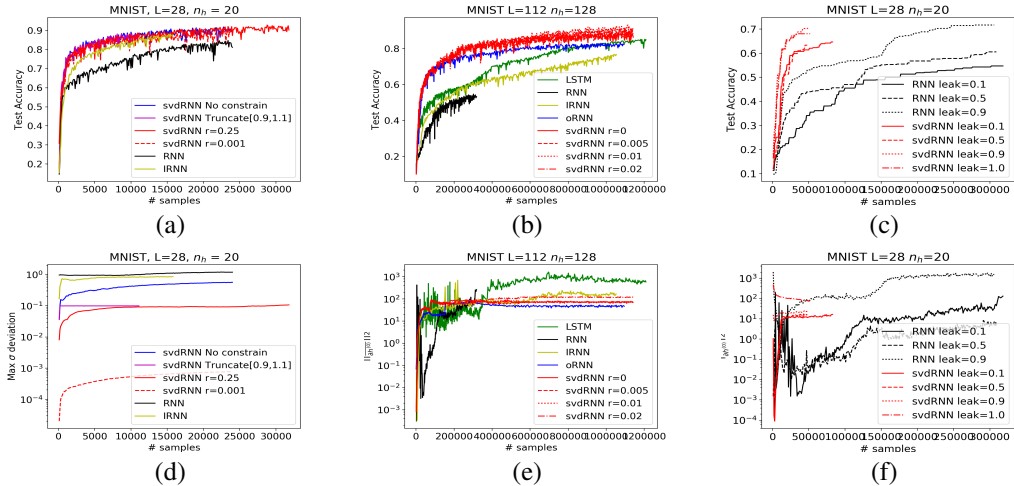

Figure 2: RNN models on MNIST

| Models | Hidden dimension | Number of parameters | Test accuracy |
|---|---|---|---|
| svdRNN | $256(m_1, m_2 = 16)$ | $\approx 13k$ | **97.6** |
| oRNN(Mhammedi et al. (2017)) | $256(m = 32)$ | $\approx 11k$ | 97.2 |
| RNN(Vorontsov et al. (2017)) | 128 | $\approx 35k$ | 94.1 |
| uRNN(Arjovsky et al. (2016)) | 512 | $\approx 16k$ | 95.1 |
| RC uRNN(Wisdom et al. (2016)) | 512 | $\approx 16k$ | 97.5 |
| FC uRNN(Wisdom et al. (2016)) | 116 | $\approx 16k$ | 92.8 |
| factorized RNN(Vorontsov et al. (2017)) | 128 | $\approx 32k$ | 94.6 |
| LSTM (Vorontsov et al. (2017)) | 128 | $\approx 64k$ | 97.3 |

Table 2: Results for the pixel MNIST dataset across multiple algorithms.

*For the MLP models, each instance is flattened to a vector of length 784 and fed to the input layer. After the input layer there are 40 layers with hidden dimension 32 (Figure 3(a)) or 30 to 100 layers with hidden dimension 128 (Figure 3(b)). On a 40-layer network, svdMLP and svdResNet achieve similar performance as ResNet while MLP's convergence is slower. However, when the network is deeper, both MLP and ResNet start to fail. With $n_h = 128$, MLP is not able to function with $L > 35$ and ResNet with $L > 70$. On the other hand, the SVD based methods are resilient to increasing depth and thus achieve higher precision.*

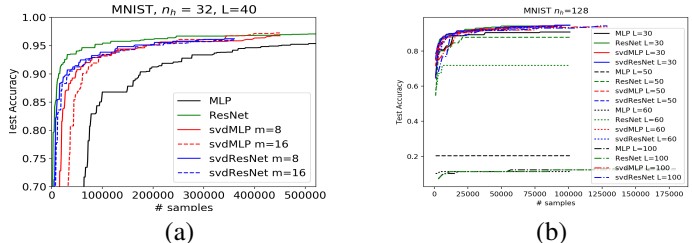

Figure 3: MLP models on MNIST with $L$ layers $n_h$ hidden dimension

## 8 CONCLUSIONS

*In this paper, we have proposed an efficient SVD parametrization of various weight matrices in deep neural networks, which allows us to explicitly track and control their singular values. This parameterization does not restrict the network's expressive power, while simultaneously allowing fast forward as well as backward propagation. The method is easy to implement and has the same time and space complexity as compared to original methods like RNN and MLP. The ability to control singular values helps in avoiding the gradient vanishing and exploding problems, and as we have empirically shown, gives good performance. Although we only showed examples in the RNN and MLP framework, our method is applicable to many more deep networks, such as Convolutional Networks etc. However, further experimentation is required to fully understand the influence of using different number of reflectors in our SVD parameterization. Also, the underlying structures of the image of $\mathcal{M}_{k_1,k_2}$ when $k_1, k_2 \neq 1$ is a subject worth investigating.*

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

## APPENDIX A    PROOFS

### A.1    PROOF OF PROPOSITION 1

**Proposition 1.** *(Householder QR factorization) Let $B \in \mathbb{R}^{n \times n}$. Then there exists an upper triangular matrix $R$ with positive diagonal elements, and vectors $\{u_i\}_{i=1}^n$ with $u_i \in \mathbb{R}^i$, such that $B = \mathcal{H}_n^n(u_n)...\mathcal{H}_1^n(u_1)R$. (Note that we allow $u_i = 0$, in which case, $H_i^n(u_i) = I_n$ as in (1))*

*Proof of Proposition 1.* For $n = 1$, note that $\mathcal{H}_1^1(u_1) = \pm 1$. By setting $u_1 = 0$ if $B_{1,1} > 0$ and $u_1 \neq 0$ otherwise, we have the factorization desired.

Assume that the result holds for $n = k$, then for $n = k + 1$ set $u_{k+1} = B_1 - \|B_1\|e_1$. Here $B_1$ is the first column of $B$ and $e_1 = (1, 0, ..., 0)^\top$. Thus we have

$$\mathcal{H}_{k+1}^{k+1}(u_{k+1})B = \begin{pmatrix} \|B_1\| & \hat{B}_{1,2:k+1} \\ 0 & \hat{B} \end{pmatrix},$$

where $\hat{B} \in \mathbb{R}^{k \times k}$. Note that $\mathcal{H}_{k+1}^{k+1}(u_{k+1}) = I_{k+1}$ when $u_{k+1} = 0$ and the above still holds. By assumption we have $\hat{B} = \mathcal{H}_k^k(u_k)...\mathcal{H}_1^k(u_1)\hat{R}$. Notice that $\mathcal{H}_i^{k+1}(u_i) = \begin{pmatrix} 1 & \\ & \mathcal{H}_i^k(u_i) \end{pmatrix}$, so we have that

$$\mathcal{H}_1^{k+1}(u_1)...\mathcal{H}_k^{k+1}(u_k)\mathcal{H}_{k+1}^{k+1}(u_{k+1})B = \begin{pmatrix} \|B_1\| & \tilde{B}_{1,2:k+1} \\ 0 & \hat{R} \end{pmatrix} = R$$

is an upper triangular matrix with positive diagonal elements. Thus the result holds for any $n$ by the theory of mathematical induction. □

### A.2    PROOF OF THEOREM 1

*Proof.* Observe that the image of $\mathcal{M}_1$ is a subset of $\mathbf{O}(n)$, and we now show that the converse is also true. Given $A \in \mathbf{O}(n)$, by Proposition 1, there exists an upper triangular matrix $R$ with positive diagonal elements, and an orthogonal matrix $Q$ expressed as $Q = \mathcal{H}_n^n(u_n)...\mathcal{H}_1^n(u_1)$ for some set of Householder vectors $\{u_i\}_{i=1}^n$, such that $A = QR$. Since $A$ is orthogonal, we have $A^\top A = AA^\top = I_n$, thus:
$$A^\top A = R^\top Q^\top QR = R^\top R = I_n; \ Q^\top AA^\top Q = Q^\top QRR^\top Q^\top Q = RR^\top = I_n$$
Thus $R$ is orthogonal and upper triangular matrix with positive diagonal elements. So $R = I_n$ and $A = Q = \mathcal{H}_n^n(u_n)...\mathcal{H}_1^n(u_1)$. □

### A.3    PROOF OF THEOREM 2

*Proof.* It is easy to see that the image of $\mathcal{M}_{1,1}$ is a subset of $\mathbb{R}^{n \times n}$. For any $W \in \mathbb{R}^{n \times n}$, we have its SVD, $W = U\Sigma V^\top$, where $\Sigma = diag(\sigma)$. By Theorem 1, for any orthogonal matrix $U, V \in \mathbb{R}^{n \times n}$, there exists $\{u_i\}_{i=1}^n\{v_i\}_{i=1}^n$ such that $U = \mathcal{M}_1(u_1, ..., u_n)$ and $V = \mathcal{M}_1(v_1, ..., v_n)$, then we have:
$$W = \mathcal{H}_n^n(u_n)...\mathcal{H}_1^n(u_1)\Sigma\mathcal{H}_1^n(v_1)...\mathcal{H}_n^n(v_n)$$
$$= \mathcal{M}_{1,1}(u_1, ..., u_n, v_1, ..., v_n, \sigma)$$
□

### A.4    PROOF OF THEOREM 3

*Proof.* Let $A \in \mathbb{R}^{n \times n}$ be an orthogonal matrix. By Theorem 1, there exist $\{a_i\}_{i=1}^n$, such that $A = \mathcal{M}_1(a_1, ..., a_n)$. Since $A^\top$ is also orthogonal, for the same reason, there exist $\{b_i\}_{i=1}^n$, such that $A^\top = \mathcal{M}_1(b_1, ..., b_n)$. Thus we have:
$$A = \mathcal{H}_n(a_n)...\mathcal{H}_1(a_1) = \mathcal{H}_1(b_1)...\mathcal{H}_n(b_n)$$
Observe that one of $k_2 \geq k_1 - 1$ and $k_1 \geq k_2 - 1$ must be true. If $k_2 \geq k_1 - 1$, set
$$u_k = a_k, k = n, n - 1, ..., k_1,$$
$$v_{k_2+k_1-k-1} = a_k, k = k_1 - 1, ..., 1, \tag{19}$$
$$v_t = \mathbf{0}, t = k_2 + k_1 - 2, ..., n,$$
and then we have:
$$\mathcal{M}_{k_1,k_2}(u_{k_1}, ..., u_n, v_{k_2}, ..., v_n, \mathbf{1}) = \mathcal{H}_n(u_n)...\mathcal{H}_{k_1}(u_{k_1})I_n\mathcal{H}_{k_2}(v_{k_2})...\mathcal{H}_n(v_n)$$
$$= \mathcal{H}_n(a_n)...\mathcal{H}_{k_1}(a_{k_1})I_n\mathcal{H}_{k_1-1}(a_{k_1-1})...\mathcal{H}_1(a_1)$$
$$= A \tag{20}$$

Else, assign:

$$v_k = b_k, \, k = n, n-1, ..., k_2,$$
$$u_{k_2+k_1-k-1} = b_k, \, k = k_2 - 1, ..., 1, \tag{21}$$
$$u_t = \mathbf{0}, \, t = k_2 + k_1 - 2, ..., n,$$

and then we have:

$$\mathcal{M}_{k_1,k_2}(u_{k_1}, ..., u_n, v_{k_2}, ..., v_n, \mathbf{1}) = \mathcal{H}_1(b_1)...\mathcal{H}_{k_2-1}(b_{k_2-1})I_n\mathcal{H}_{k_2}(b_{k_2})...\mathcal{H}_n(b_n)$$
$$= A \tag{22}$$

$\square$

## A.5 Proof of Theorem 4

*Proof.* It is easy to see that the image of $\mathcal{M}_{*,*}^{m,n}$ is a subset of $\mathbb{R}^{m \times n}$. For any $W \in \mathbb{R}^{m \times n}$, we have its SVD, $W = U\Sigma V^\top$, where $\Sigma$ is an $m \times n$ diagonal matrix. By Theorem 1, for any orthogonal matrix $U \in \mathbb{R}^{m \times m}, V \in \mathbb{R}^{n \times n}$, there exists $\{u_i\}_{i=1}^m \{v_i\}_{i=1}^n$ such that $U = \mathcal{H}_m^m(u_m)...\mathcal{H}_1^m(u_1)$ and $V = \mathcal{H}_n^n(v_n)...\mathcal{H}_1^n(v_1)$. By Lemma 1, if $m < n$ we have:

$$W = \mathcal{H}_n^m(u_n)...\mathcal{H}_1^m(u_1)\Sigma\mathcal{H}_1^n(v_1)...\mathcal{H}_n^n(v_n)$$
$$= \mathcal{H}_n^m(u_n)...\mathcal{H}_1^m(u_1)\Sigma\mathcal{H}_{n-m+1}^n(v_{n-m+1})...\mathcal{H}_n^n(v_n).$$

Similarly, for $n < m$, we have:

$$W = \mathcal{H}_n^m(u_n)...\mathcal{H}_1^m(u_1)\Sigma\mathcal{H}_1^n(v_1)...\mathcal{H}_n^n(v_n)$$
$$= \mathcal{H}_n^m(u_n)...\mathcal{H}_{m-n+1}^m(u_{m-n+1})\Sigma\mathcal{H}_1^n(v_1)...\mathcal{H}_n^n(v_n).$$

$\square$

## A.6 Proof of Claim 1

*Proof.*
$$R[W] = \mathbb{E}_{\mathcal{X} \sim \mathcal{D}}[f(W; \mathcal{X}, y)]$$
$$= \mathbb{E}_{\mathcal{X} \sim \mathcal{D}}\left[\|(Y\mathcal{W} - A)\text{vec}(\mathcal{X}) - \eta\|^2\right]$$
$$= \mathbb{E}\left[tr\left((Y\mathcal{W} - A)\text{vec}(\mathcal{X})\text{vec}(\mathcal{X})^\top(Y\mathcal{W} - A)^\top\right)\right] + \mathbb{E}[\|\eta\|^2] - 2\mathbb{E}\left[\langle(Y\mathcal{W} - A)\text{vec}(\mathcal{X}), \eta\rangle\right]$$
$$= tr\left((Y\mathcal{W} - A)\mathbb{E}\left[\text{vec}(\mathcal{X})\text{vec}(\mathcal{X})^\top\right](Y\mathcal{W} - A)^\top\right) + C$$
$$= \|(Y\mathcal{W} - A)\Sigma^{1/2}\|_F^2 + C$$

For the derivative,
$$R[W + \Delta W] = \|(Y\left(W + \Delta W|(W + \Delta W)^2|\cdots|(W + \Delta W)^t\right) - A)\Sigma^{1/2}\|_F^2 + C$$
$$= R[W] + \left\langle\left(\Delta W|2W\Delta W|\cdots|tW^{t-1}\Delta W\right)\Sigma^{1/2}, (Y\mathcal{W} - A)\Sigma^{1/2}\right\rangle + o(\|\Delta W\|_F^2)$$
$$= R[W] + \left\langle\Delta W, (Y\mathcal{W} - A)\Sigma\left(I_d|2W|3W^2|\cdots|tW^{t-1}\right)^\top\right\rangle$$

Therefore $\nabla R[W] = (Y\mathcal{W} - A)\Sigma\left(I_d|2W|3W^2|\cdots|tW^{t-1}\right)^\top$

Remark: here when $W$ and $\Delta W$ are not commutative, each $W^i\Delta W$ should instead be written as $\sum_{j=0}^i W^j\Delta WW^{i-j}$. Since the change of order doesn't impact the analysis, we informally simplify the expressions here.

$\square$

## APPENDIX B    DETAILS OF FORWARD AND BACKWARD PROPAGATION ALGORITHMS

---

**Algorithm 1** Local forward/backward propagation

---

**Input**: $h^{(t-1)}, \frac{\partial L}{\partial C^{(t)}}, U = (u_n|...|u_{n-m_1+1})$,
$\Sigma, V = (v_n|...|v_{n-m_2+1})$
**Output**: $C^{(t)} = Wh^{(t-1)}, \frac{\partial L}{\partial U}, \frac{\partial L}{\partial V}, \frac{\partial L}{\partial \hat{\sigma}}, \frac{\partial L}{\partial h^{(t-1)}}$
// Begin forward propagation
$h_{n+1}^{(v)} \leftarrow h^{(t-1)}$
**for** $k = n, n-1, ..., n-m_2+1$ **do**
    $h_k^{(v)} \leftarrow Hprod(h_{k+1}^{(v)}, v_k)$    // Compute $\hat{V}^\top h$
**end for**
$h_{k_1-1}^{(u)} \leftarrow \Sigma h_{k_2}^{(v)}$          // Compute $\Sigma \hat{V}^\top h$
**for** $k = n-m_1+1, ..., n$ **do**
    $h_k^{(u)} \leftarrow Hprod(h_{k-1}^{(u)}, u_k)$    // Compute $\hat{U}\Sigma\hat{V}^\top h$
**end for**
$C^{(t)} \leftarrow h_n^{(u)}$
//Begin backward propagation
$g \leftarrow \frac{\partial L}{\partial C^{(t)}}$
**for** $k = n, n-1, ..., n-m_1+1$ **do**
    $g, G_{*,n-k+1}^{(u)} \leftarrow Hgrad(h_k^{(u)}, u_k, g)$    // Compute $\frac{\partial L}{\partial u_k}$
**end for**
$\bar{\Sigma} \leftarrow diag(g \circ h_{k_2}^{(v)}), g \leftarrow \Sigma g$          // Compute $\frac{\partial L}{\partial \Sigma}$
$g^{(\hat{\sigma})} \leftarrow \frac{\partial diag(\Sigma)}{\partial \hat{\sigma}} \circ diag(\bar{\Sigma})$          // Compute $\frac{\partial L}{\partial \hat{\sigma}}$
**for** $k = n-m_2+1, ..., n$ **do**
    $g, G_{*,n-k+1}^{(v)} \leftarrow Hgrad(h_{k+1}^{(u)}, v_k, g)$    // Compute $\frac{\partial L}{\partial v_k}$
**end for**
$\frac{\partial L}{\partial U} \leftarrow G^{(u)}, \frac{\partial L}{\partial V} \leftarrow G^{(v)}, \frac{\partial L}{\partial \hat{\sigma}} \leftarrow g^{(\hat{\sigma})}, \frac{\partial L}{\partial h^{(t-1)}} \leftarrow g$

---

**Algorithm 2**

---

$\hat{h} = Hprod(h, u_k)$

---

**Input**: $h, u_k$
**Output**: $\hat{h} = \mathcal{H}_k(u_k)h$
// Compute $\hat{h} = (I - \frac{2u_k u_k^\top}{u_k^\top u_k})h$
$\alpha \leftarrow \frac{2}{\|u_k\|^2} u_k^\top h$
$\hat{h} \leftarrow h - \alpha u_k$

---

**Algorithm 3**

---

$\bar{h}, \bar{u}_k = Hgrad(h, u_k, g)$

---

**Input**: $h, u_k, g = \frac{\partial L}{\partial C}$ where $C = \mathcal{H}_k(u_k)h$
**Output**: $\bar{h} = \frac{\partial L}{\partial h}, \bar{u}_k = \frac{\partial L}{\partial u_k}$
$\alpha = \frac{2}{\|u_k\|^2} u_k^\top h$
$\beta = \frac{2}{\|u_k\|^2} u_k^\top g$
$\bar{h} \leftarrow g - \beta u_k$
$\bar{u}_k \leftarrow -\alpha g - \beta h + \alpha\beta u_k$

---

## APPENDIX C    MORE EXPERIMENTAL DETAILS

### C.1    DETAILS ON THE TIME SERIES CLASSIFICATION TASK

*For the time series classification task, we use the training and testing sets directly from the UCR time series archive* `http://www.cs.ucr.edu/~eamonn/time_series_data/`*, and randomly choose 20% of the training set as validation data. We provide the statistical descriptions of the datasets and experimental results in Table 3.*

| Datasets | Data Descriptions | | | | Depth | RNN | | LSTM | | oRNN | | svdRNN | |
|---|---|---|---|---|---|---|---|---|---|---|---|---|---|
| | training/testing size | length | #class | | | acc ($n_{param}$) | | acc ($n_{param}$) | | acc ($n_{param}$) | | acc ($n_{param}$) | |
| 50words | 450 | 455 | 270 | 50 | 27 | 0.492 | (3058) | 0.598 | (7218) | 0.642 | (2426) | **0.651** | (2850) |
| Adiac | 390 | 391 | 176 | 37 | 16 | 0.552 | (2694) | 0.706 | (6950) | 0.668 | (2062) | **0.726** | (2486) |
| ArrowHead | 36 | 175 | 251 | 3 | 251 | 0.509 | (1219) | 0.537 | (4515) | 0.669 | (587) | **0.800** | (1011) |
| Beef | 30 | 30 | 470 | 5 | 47 | 0.600 | (1606) | 0.700 | (5766) | **0.733** | (974) | **0.733** | (1398) |
| BeetleFly | 20 | 20 | 512 | 2 | 32 | **0.950** | (1699) | 0.850 | (6435) | 0.900 | (1067) | **0.950** | (1491) |
| CBF | 30 | 900 | 128 | 3 | 16 | 0.702 | (1476) | **0.967** | (5444) | 0.881 | (844) | 0.948 | (1268) |
| Coffee | 28 | 28 | 286 | 2 | 22 | **1.000** | (1570) | **1.000** | (6018) | **1.000** | (938) | **1.000** | (1362) |
| Cricket X | 390 | 390 | 300 | 12 | 20 | 0.310 | (1997) | 0.456 | (6637) | 0.495 | (1365) | **0.500** | (1789) |
| DistalPhalanxOutlineCorrect | 276 | 600 | 80 | 2 | 10 | 0.790 | (1410) | 0.798 | (5378) | 0.830 | (778) | **0.840** | (1202) |
| DistalPhalanxTW | 154 | 399 | 80 | 6 | 10 | **0.815** | (1641) | 0.795 | (5609) | 0.807 | (1009) | **0.815** | (1433) |
| ECG200 | 100 | 100 | 96 | 2 | 12 | **0.640** | (1410) | **0.640** | (5378) | **0.640** | (778) | **0.640** | (1202) |
| ECG5000 | 500 | 4500 | 140 | 5 | 14 | 0.941 | (1606) | 0.936 | (5766) | 0.940 | (974) | **0.945** | (1398) |
| ECGFiveDays | 23 | 861 | 136 | 2 | 17 | 0.947 | (1443) | 0.790 | (5411) | **0.976** | (811) | 0.948 | (1235) |
| FaceAll | 560 | 1690 | 131 | 14 | 131 | 0.549 | (1615) | 0.455 | (4911) | **0.714** | (983) | **0.714** | (1407) |
| FaceFour | 24 | 88 | 350 | 4 | 25 | 0.625 | (1701) | 0.477 | (6245) | 0.511 | (1069) | **0.716** | (1493) |
| FacesUCR | 200 | 2050 | 131 | 14 | 131 | 0.449 | (1615) | 0.629 | (4911) | 0.710 | (983) | **0.727** | (1407) |
| Gun Point | 50 | 150 | 150 | 2 | 15 | 0.947 | (1507) | 0.920 | (5667) | 0.953 | (875) | **0.960** | (1299) |
| InsectWingbeatSound | 220 | 1980 | 256 | 11 | 16 | 0.534 | (1996) | 0.515 | (6732) | **0.598** | (1364) | 0.586 | (1788) |
| ItalyPowerDemand | 67 | 1029 | 24 | 2 | 6 | 0.970 | (1315) | 0.969 | (4899) | 0.972 | (683) | **0.973** | (1107) |
| Lighting2 | 60 | 61 | 637 | 2 | 49 | **0.541** | (1570) | **0.541** | (6018) | **0.541** | (938) | **0.541** | (1362) |
| MiddlePhalanxOutlineCorrect | 291 | 600 | 80 | 2 | 10 | 0.793 | (1410) | 0.783 | (5378) | 0.712 | (778) | **0.820** | (1202) |

Table 3: Test accuracy (number of parameters) on UCR datasets. For each dataset, we present the testing accuracy when reaching the smallest validation error. The highest precision is in bold, and lowest two are colored gray.

## C.2 EXPERIMENTAL RESULTS ON THE ADDING AND COPYING TASKS

*We tested RNN models on the Adding and Copying tasks with the same settings as Arjovsky et al. (2016).*

### C.2.1 ADDING TASK

*The Adding task requires the network to remember two marked numbers in a long sequence and add them. Each input data includes two sequences: top sequence whose values are sampled uniformly from $[0, 1]$ and bottom sequence which is a binary sequence with only two 1's. The network is asked to output the sum of the two values. From the empirical results in Figure 4, we can see that when the network is not deep (number of layers L=30 in (a)(d)), every model outperforms the baseline of 0.167 (always output 1 regardless of the input). Also, the first layer gradients do not vanish for all models. However, on longer sequences (L=100 in (b)(e)), IRNN failed and LSTM converges much slower than svdRNN and oRNN. If we further increase the sequence length (L=300 in (c)(f)), only svdRNN and oRNN are able to beat the baseline within reasonable number of iterations. We can also observe that the first layer gradient of oRNN/svdRNN does not vanish regardless of the depth, while IRNN/LSTM's gradient vanish as L becomes lager.*

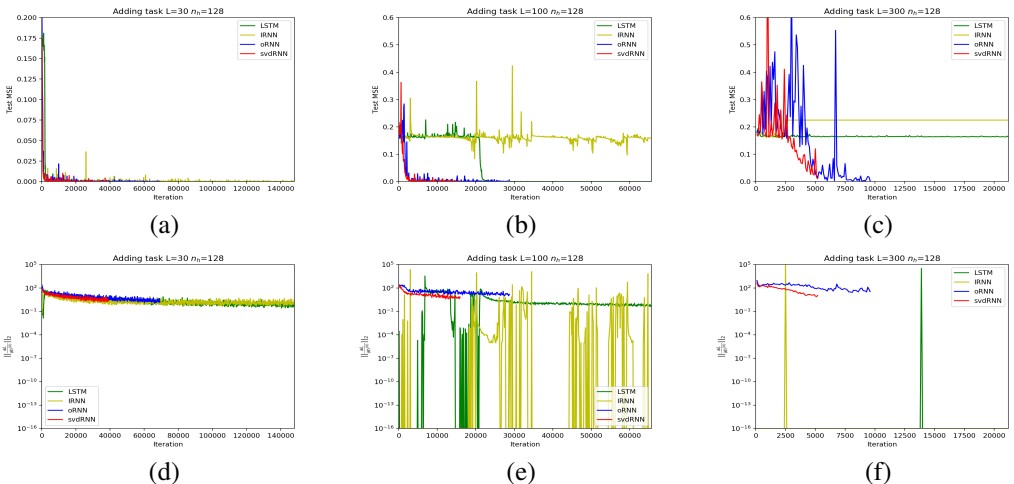

Figure 4: RNN models on the adding task with $L$ layers and $n_h$ hidden dimension. The top plots show the test MSE, while the bottom plots show the magnitude of the gradient at the first layer.

### C.2.2 COPYING TASK

*Let $A = \{a_i\}_{i=0}^9$ be the alphabet. The input data sequence $x \in A^{T+20}$ where $T$ is the time lag. $x_{1:10}$ are sampled uniformly from $i\{a_i\}_{i=0}^7$ and $x_{T+10}$ is set to $a_9$. Rest of $x_i$ is set to $a_8$. The network is asked to output $x_{1:10}$ after seeing $a_9$. That is to copy $x_{1:10}$ from the beginning to the end with time lag $T$.*

*A baseline strategy is to predict $a_8$ for $T+10$ entrees and randomly sample from $\{a_i\}_{i=1}^7$ for the last 10 digits. From the empirical results in Figure 5, svdRNN consistently outperforms all other models. IRNN and LSTM models are not able to beat the baseline with large time lag. In fact, the loss of RNN/LSTM is very close to the baseline (memoryless strategy) indicates that they do not memorize any useful information throughout the time lag.*

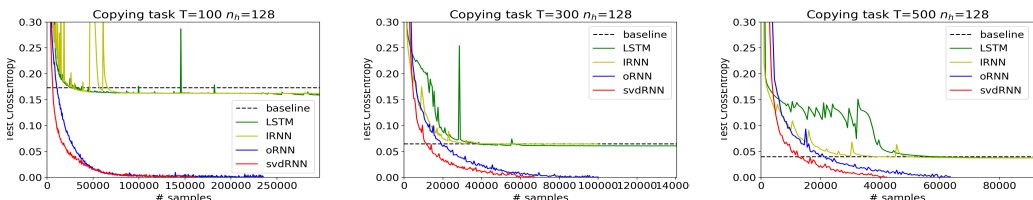

Figure 5: RNN models on the Copying task with $T$ time lag and $n_h$ hidden dimension.

