# OpenReview forum: "Stabilizing Gradients for Deep Neural Networks via Efficient SVD Parameterization"
_ICLR.cc/2018/Conference — Reject_

### Official Review · AnonReviewer2 · 2017-11-26

**Rating:** 7
**Confidence:** 4

**Review:**

This paper proposed a new parametrization scheme for weight matrices in neural network based on the Householder  reflectors to solve the gradient vanishing and exploding problems in training. The proposed method improved two previous papers:
1) stronger expressive power than Mahammedi et al. (2017),
2) faster gradient update than Vorontsov et al. (2017).
The proposed parametrization scheme is natrual from numerical linear algebra point of view and authors did a good job in Section 3 in explaining the corresponding expressive power. The experimental results also look promising.

It would be nice if the authors can analyze the spectral properties of the saddle points in linear RNN (nonlinear is better but it's too difficult I believe). If the authors can show the strict saddle properties then as a corollary, (stochastic) gradient descent finds a global minimum.

Overall this is a strong paper and I recommend to accept.

---

> ### Author Response · Authors · 2017-12-29
> **Rebuttal 2**
>
> Thank you for your insightful comments. We'll consider your suggestions and make modifications to the paper.
>
> It is certainly important for the community to study the spectral properties of different types of stationary points of linear RNN. Nevertheless in our work we focus on how our proposed svdRNN helps to avoid local minimum, which is guaranteed by theorem 5 and its corollary.

---

### Official Review · AnonReviewer1 · 2017-11-28
**SVD reparametrization of the transition matrix**

**Rating:** 5
**Confidence:** 4

**Review:**

This paper suggests a reparametrization of the transition matrix. The proposed reparametrization which is based on Singular Value Decomposition can be used for both recurrent and feedforward networks.

The paper is well-written and authors explain related work adequately. The paper is a follow up on Unitary RNNs which suggest a reparametrization that forces the transition matrix to be unitary. The problem of vanishing and exploding gradient in deep network is very challenging and any work that shed lights on this problem can have a significant impact.

I have two comments on the experiment section:

- Choice of experiments. Authors have chosen UCR datasets and MNIST for the experiments while other experiments are more common. For example, the adding problem, the copying problem and the permuted MNIST problem and language modeling are the common experiments in the context of RNNs. For feedforward settings, classification on CIFAR10 and CIFAR100 is often reported.

- Stopping condition. The plots suggest that the optimization has stopped earlier for some models. Is this because of some stopping condition or because of gradient explosion? Is there a way to avoid this?

- Quality of figures. Figures are very hard to read because of small font. Also, the captions need to describe more details about the figures.

---

> ### Author Response · Authors · 2017-12-29
> **Rebuttal 1**
>
> Thank you for your insightful comments. We have carefully reorganized the paper to take into account your suggestions.
>
> We actually have done experiments on Adding and Copying tasks -- these classical benchmarking tasks can test the model's ability in learning long term dependencies. In the new version, we have included experimental result of Adding/Copying problem in Appendix C. On short sequnences, all models performs similarly well. However, svdRNN outperforms other models when sequence length is large.
> For example, on the addition task, when length of the sequence is 300, svdRNN reached almost 0 loss after 5k examples while RNN/LSTM failed to converge within 20k examples. From the plot of first layer gradient magnitude on different models, we can observe that svdRNN has much more stable gradients than RNN/LSTM.
> These experiments support our claim that svdRNN is much better at capturing long-term dependencies than vanilla RNNs.
>
> In most experiments we run all models for the same number of iterations. In the experiment on MNIST, RNN with random initialization encountered gradient explosion and stopped early. Implementing gradient clipping might be able to solve this. However, to have a fair comparison among the algorithms, we did not use huristics like gradient clipping on any of these models.
>
> Sorry for the small font in figures. We have enlarged the fonts in figures, and added more details in the figure captions.

---

### Official Review · AnonReviewer3 · 2017-12-02

**Rating:** 5
**Confidence:** 3

**Review:**

The paper introduces SVD parameterization and uses it mostly for controlling the spectral norm of the RNN.

My concerns with the paper include:

a) the paper says that the same method works for convolutional neural networks but I couldn't find anything about convolution.

b) the theoretical analysis might be misleading --- clearly section 6.2 shouldn't have title ALL CRITICAL POINTS ARE GLOBAL MINIMUM because 0 is a critical point but it's not a global minimum. Theorem 5 should be phrased as

all critical points of the population risk that is non-singular are global minima.

c) the paper should run some experiments on language applications where RNN is widely used

d) I might be wrong on this point, but it seems that the GPU utilization of the method would be very poor so that it's kind of impossible to scale to large datasets?

---

> ### Author Response · Authors · 2017-12-29
> **Rebuttal 3**
>
> Thank you for your comments. We have carefully reorganized the paper to take into account your suggestions.
>
> Since our parameterization works for any real matrix, any weight matrix in CNN (e.g. filter matrix) can also be SVD parametrized.
> Since the gradient exploding/vanishing problem is generally considered more significant in RNN/MLP models than in CNN, we have not implement our algorithm as yet for CNNs.
>
> (Please also note that the state-of-the-art neural networks for image processing consists of both convolutional layers and fully-connected layers, like LeNet, AlexNet, ResNet, or Inception, etc. We are already able to parameterize the fully connected layers using our scheme. )
>
> We will explicitly explain in section 6.2 that we are talking about non-singular critical points. Our theory effectively states that our update rule avoids singular critical points, and thus spurious local minimum.
>
> In Appendix C (experimental results), we have added experiments on widely used benchmarking RNN tasks like Adding and Copying tasks. In all experiments, svdRNN outperforms others expecially in capturing long-range dependencies. Also from the plots of (first layer) gradient magnitude, we can observe that svdRNN has much more stable gradient than that of RNN and LSTM, which could be the reason why it converges much faster.
>
> The efficiency and GPU utilization is indeed an important aspect.
> However, with efficient linear algebra algorithms, GPU utilization of our method can actually be quite high. The main part in our algorithm, where we multiply m Householder reflectors to the hidden matrix (stacking of hidden vector h's within a minibatch), can be done using blocked (BLAS3) algorithm widely used in QR decomposition.
> For example, in LAPACK library, the corresponding subroutine is called 'DLARFT', while in MAGMA (LAPACK with GPU) it is called 'magma_cunmqr'. We plan to implement these blocked BLAS3 algorithms as part of our software in the near future.

---

### Decision · Program_Chairs · 2018-01-29
**ICLR 2018 Conference Acceptance Decision**

**Decision:**

Reject

**Comment:**

Pros:
+ Clearly written paper.
+ Good theoretical analysis of the expressivity of the proposed model.
+ Efficient model update is appealing.
+ Reviewers appreciated the addition of results on the copy and adding tasks in Appendix C.

Cons:
- Evaluation was on less-standard RNN tasks.  A language modeling task should have been included in the empirical evaluation because language modeling is such an important application of RNNs.

This paper is close to the decision boundary, but the reviewers strongly felt that demonstration of the method on a language modeling task was necessary for acceptance.